# Thioredoxin System Protein Expression in Carcinomas of the Pancreas, Distal Bile Duct, and Ampulla in the United Kingdom

**DOI:** 10.3390/diseases12100227

**Published:** 2024-09-24

**Authors:** Khaled S. Al-Hadyan, Sarah J. Storr, Abed M. Zaitoun, Dileep N. Lobo, Stewart G. Martin

**Affiliations:** 1Nottingham Breast Cancer Research Centre, School of Medicine, Biodiscovery Institute, University of Nottingham, University Park, Nottingham NG7 2RD, UK; khadyan@kfshrc.edu.sa (K.S.A.-H.); sarah.storr@nottingham.ac.uk (S.J.S.); 2Radiation Biology Section, Biomedical Physics Department, King Faisal Specialist Hospital and Research Centre, Riyadh 11211, Saudi Arabia; 3Department of Cellular Pathology, Nottingham University Hospitals NHS Trust, Queen’s Medical Centre, Nottingham NG7 2UH, UK; abd.zaitoun@nuh.nhs.uk; 4National Institute for Health Research Nottingham Biomedical Research Centre, Nottingham University Hospitals NHS Trust and University of Nottingham, Queen’s Medical Centre, Nottingham NG7 2RD, UK; 5MRC Versus Arthritis Centre for Musculoskeletal Ageing Research, School of Life Sciences, University of Nottingham, Queen’s Medical Centre, Nottingham NG7 2UH, UK; dileep.lobo@nottingham.ac.uk

**Keywords:** thioredoxin, redox protein, periampullary cancer, pancreatic ductal adenocarcinoma, ampullary cancer, bile duct cancer

## Abstract

**Background:** Poor survival outcomes in periampullary cancer highlight the need for improvement in biomarkers and the development of novel therapies. Redox proteins, including the thioredoxin system, play vital roles in cellular antioxidant systems. **Methods:** In this retrospective study, thioredoxin (Trx), thioredoxin-interacting protein (TxNIP), and thioredoxin reductase (TrxR) protein expression was assessed in 85 patients with pancreatic ductal adenocarcinoma (PDAC) and 145 patients with distal bile duct or ampullary carcinoma using conventional immunohistochemistry. **Results:** In patients with PDAC, high cytoplasmic TrxR expression was significantly associated with lymph node metastasis (*p* = 0.033). High cytoplasmic and nuclear Trx expression was significantly associated with better overall survival (*p* = 0.018 and *p* = 0.006, respectively), and nuclear Trx expression remained significant in multivariate Cox regression analysis (*p* < 0.0001). In distal bile duct and ampullary carcinomas, high nuclear TrxR expression was associated with vascular (*p* = 0.001) and perineural (*p* = 0.021) invasion, and low cytoplasmic TxNIP expression was associated with perineural invasion (*p* = 0.025). High cytoplasmic TxNIP expression was significantly associated with better overall survival (*p* = 0.0002), which remained significant in multivariate Cox regression analysis (*p* = 0.013). **Conclusions:** These findings demonstrate the prognostic importance of Trx system protein expression in periampullary cancers.

## 1. Introduction

Pancreatic cancer (PC) is the seventh leading cause of cancer-related deaths worldwide in both sexes, accounting for 466,000 deaths in 2020 [1]. Five-year survival rates have been reported to be better in patients with distal bile duct and ampullary cancers than in those with pancreatic ductal adenocarcinoma (PDAC) (29–37% and 41–45% vs. 7.8%, respectively) [2,3,4,5]. Over the last 30–40 years, treatment modalities for PDAC, including surgery, chemotherapy, and radiotherapy, have failed to improve the five-year survival rate, which remains low at ≤7.8% in the United Kingdom [2], highlighting the urgent need to identify new molecular targets for the development of novel therapeutic approaches.

The thioredoxin (Trx) system, which includes Trx, the activating enzyme thioredoxin reductase (TrxR), and the endogenous inhibitor thioredoxin-interacting protein (TxNIP), belongs to a large family of redox proteins. As such, the Trx system is involved in the regulation of redox homeostasis and affects the redox state of a range of signaling proteins, thereby controlling numerous downstream pathways involved in the regulation of cell growth, apoptosis, gene transcription, cell cycle progression, and oxidative stress [6,7,8,9,10,11,12]. Therefore, inhibiting antioxidant pathways is a potentially powerful approach to selectively target transformed cells, including PC cells [6,7,8,9,10,11,12]. However, therapeutic targeting of the antioxidant system may be more challenging in PC than in other cancers [13,14]. PC cells exhibit increased autophagy, a cellular process that aids in maintaining appropriate levels of reactive oxygen species (ROS) and consequently supports mitochondrial metabolism [13,14]. Given that the Trx system plays a major role in maintaining intracellular ROS levels in cancer cells, including periampullary cancer cells, inhibiting the Trx system might alter the intracellular redox state and induce apoptosis, thereby rendering cancer cells more sensitive to treatments such as chemotherapy and radiotherapy [15].

Trx proteins (12-kDa proteins) include cytosolic thioredoxin-1 (Trx1), mitochondrial thioredoxin-2 (Trx2), and thioredoxin-like protein. Trx1 is an extensively studied cytosolic protein with disulfide reductase activity [16]. Reduced Trx1, the bioactive form of Trx1, binds to apoptosis signal-regulating kinase 1, a key apoptotic regulator whose activation is essential for tumor necrosis factor α-induced apoptosis, and inactivates it, thereby providing protection against apoptosis [17]. The role of the Trx system in cell growth and apoptosis can also be explained by the selective activation of a number of transcription factors such as nuclear factor kappa-light-chain-enhancer of activated B cells, glucocorticoid receptor, p53, activator protein 1, and activator protein 2 [16]. Few studies have investigated the mechanism, function, and effect of Trx2 in cancer treatment. However, Trx2 has been extensively demonstrated to maintain cardiac function by eliminating ROS, preserving mitochondrial integrity, and suppressing apoptosis [18,19]. Importantly, in one study aimed to elucidate the roles of the Trx proteins in different compartments of human colonic epithelial cells, the measurement of protein levels under oxidative stress caused by limited energy supply revealed that the nuclear Trx1 provided better protection against oxidative stress than the cytosolic Trx1 or the mitochondrial Trx2 [20].

TrxR is a homodimeric pyridine nucleotide–disulfide oxidoreductase localized in the cytosol and mitochondria [21]. TrxR is the only known enzyme that can reduce oxidized Trx by catalyzing the transfer of electrons from NADPH to oxidized Trx [22]. Therefore, TrxR plays an important role in regulating cell growth, proliferation, apoptosis, and redox homeostasis, and its inhibition induces Trx oxidation, leading to the activation of the p38 and JNK signaling pathway and downstream apoptosis [23].

TxNIP, also known as vitamin D3 upregulated protein 1 or Trx-binding protein 2, is a stress-responsive protein that inhibits Trx activity by preventing the recycling of oxidized Trx to its reduced form [21]. TxNIP can inhibit the activity of Trx through two pathways. First, oxidized TxNIP (Cys247) binds reduced Trx (Cys32) and acts as a competitive inhibitor to remove Trx from proteins such as apoptosis signal-regulating kinase 1, thereby inhibiting their functions. Second, upregulation of TxNIP expression by factors such as disturbed flow and high glucose concentration results in decreased TrxR activity, leading to an increase in oxidative stress and apoptosis [24].

Numerous studies have investigated the relationship between redox proteins, including the Trx system proteins, and clinicopathological features and survival in patients with various cancers, including ovarian, breast, gastroesophageal, colorectal, and brain cancers [25,26,27,28,29,30,31]. However, only two studies have evaluated the expression levels of Trx system proteins in PC, reporting the absence of an association with clinicopathological features and survival outcomes [30,31]. The first study indicated a potential association between Trx and glutaredoxin proteins with malignancy potential in a small cohort of 32 patients with PDAC. The authors reported that the rate of immunohistochemical Trx positivity was higher in PDAC tissue (75%) than in pancreatic cystadenocarcinoma or normal pancreatic tissue [32]. The second study found that TxNIP was not expressed in 22 of 36 patients with PDAC [33].

Several studies have evaluated the expression of Trx-related proteins, such as peroxiredoxin (Prx) proteins in PC. In a recent study evaluating the expression of five Prx isoforms (Prx I, II, III, V, and VI) in pretreatment samples of 69 patients with PDAC [34], higher Prx I expression was significantly associated with longer relapse-free survival (*p* = 0.041), whereas higher Prx VI expression was significantly associated with longer disease-free survival (*p* = 0.0037) in patients with stage T3–4 tumors. Furthermore, stronger cytoplasmic Prx III expression was significantly associated with node-negative status (*p* = 0.007) and better tumor differentiation (*p* = 0.033), whereas higher cytoplasmic Prx V expression was significantly associated with smaller tumor size (*p* = 0.029) and node-negative status (*p* = 0.003) [34]. Recently, another study confirmed these findings by reporting that higher nuclear Prx I expression was associated with longer survival in a cohort of 60 patients with PC (*p* = 0.001) [31].

The present study aimed to determine whether Trx, TrxR, and TxNIP were expressed in carcinomas of the pancreas, distal bile duct, and ampulla and whether their expression was associated with clinicopathological features and patient survival.

## 2. Materials and Methods

### 2.1. Clinical Samples

The expression levels of the Trx system proteins were determined using tissue microarrays of tumor tissues collected from patients treated at Nottingham University Hospitals between 1993 and 2013 according to REMARK criteria [35]. The REMARK checklist includes 20 items utilized for reporting studies on prognostic tumor markers by providing positive examples and empirical evidence of the quality of reporting [35]. Ethics approval was obtained from the Health Research Authority (approval no. 18/HRA/0292) for the use of anonymized archival specimens, and the requirement for patient/relative consent was waived by the Ethics Committee. The overall cohort comprised 230 patients, including 85 patients with PDAC and 145 patients with distal bile duct (*n* = 80) or ampullary (n = 65) carcinoma. In brief, 21 of the 85 patients (61%) with PDAC were male, with an age range of 35–81 years and a median age of 66 years, whereas 81 of the 145 patients (56%) with distal bile duct or ampullary carcinoma were male, with an age range of 39–85 years and a median age of 65 years. The age cutoff of 60 was selected based on established data linking increased PC risk to individuals aged 60 and older, with only a small fraction of cases occurring before that age [36,37]. At the time of admission, information on ethnicity was not systematically collected or analyzed. PC staging was classified according to the TNM staging system as follows: the size of the primary pancreatic tumor (T), nearby lymph node involvement (N), and distant metastases (M) [38]. Table 1 shows the clinicopathological characteristics of the study cohort.

In this study, survival was calculated from the date of surgery to the date of death or to the last date of confirmation of life for censored patients. The median survival time was 18.2 months for the PDAC cohort and 18.0 months for the distal bile duct/ampullary carcinoma cohort.

In the PDAC cohort, among the 67 patients with available data, 36 (53.7%) received adjuvant chemotherapy, including fluorouracil/folinic acid and gemcitabine chemotherapy in 19 (52.8%) and 5 (13.8%) patients, respectively. In the distal bile duct/ampullary carcinoma cohort, among the 95 patients with available data, 27 (28.7%) received adjuvant chemotherapy, including fluorouracil/folinic acid in 13 (48.1%) patients and gemcitabine chemotherapy in 5 (18.5%) patients.

### 2.2. Western Blotting

The specificity of primary antibodies against Trx, TrxR, and TxNIP to be used for immunohistochemical analysis was initially assessed via Western blotting of lysates prepared from PANC-1, MIA PaC-2, and BxPc-3 PC cell lines; MDA-MB-231 and MCF-7 breast cancer cell lines were used as positive controls. All cell lines were originally obtained from the American Type Culture Collection, with authentication conducted every 4–6 months using short tandem repeat profiling. Lysates of subconfluent cells were prepared by resuspending them in radio-immunoprecipitation assay buffer (Sigma, Hertfordshire, UK) supplemented with protease and phosphatase inhibitor cocktails (Thermo Fisher Scientific, Cheshire, UK) and ethylenediaminetetraacetic acid (Thermo Fisher Scientific, Cheshire, UK). Proteins were separated via sodium dodecyl sulfate–polyacrylamide gel electrophoresis (SDS–PAGE) and transferred onto 0.2 μm nitrocellulose membranes. After blocking with 5% (*w*/*v*) milk powder in phosphate-buffered saline with 0.1% Tween, the membranes were incubated with primary antibodies overnight at 4 °C. The following primary antibodies from Abcam (Cambridge, UK) were used: rabbit anti-human Trx antibody (1:5000 dilution; Ab133524), mouse anti-human TrxR antibody (1:1000 dilution; Ab16847), and rabbit anti-human TxNIP antibody (1:1000 dilution; Ab188865). Mouse anti-human β-actin antibody was used as the loading control (1:2000 dilution; Ab8226). After incubation with primary antibodies, the membranes were incubated with anti-mouse or anti-rabbit horseradish peroxidase-conjugated secondary antibodies from Dako (Santa Clara, CA, USA) for 1 h at room temperature, and the protein bands were visualized using an ECL reagent (Amersham, UK) with Hyperfilm (GE Healthcare, Buckinghamshire, UK).

### 2.3. Tissue Microarray and Immunohistochemistry

Antibody concentrations for immunohistochemistry were optimized using three full-face tissue sections from three PDAC, three distal bile duct adenocarcinoma, and three ampullary adenocarcinoma samples.

Tissue microarray construction and immunohistochemical analyses were previously described [39]. In brief, slides with tissue cores were initially deparaffinized in xylene and then rehydrated in ethanol and water. Antigen retrieval was performed by heating the slides in 0.01 mol/L sodium citrate buffer (pH 6.0) using a microwave at 750 W for 10 min and at 450 W for 10 min. Tissue cores were treated with peroxidase block, washed with tris-buffered saline, and treated with a protein block solution. Primary anti-Trx, anti-TrxR, and anti-TxNIP antibodies were diluted at 1:2000, 1:100, and 1:250, respectively, and the tissue cores were incubated with primary anti-Trx and anti-TxNIP antibodies for 1 h at room temperature or with primary anti-TrxR antibody overnight at 4 °C. Next, the tissue cores were washed with tris-buffered saline prior to the application of post-primary solution, followed by the application of Novolink polymer solution. An immunohistochemical reaction was developed using 3,3′-diaminobenzidine as the chromogenic substrate, and the tissue cores were counterstained with hematoxylin prior to dehydration in ethanol and fixation in xylene. Composite breast tumor cores, including six stage 1 breast tumors with grades ranging from 1 to 3, were included as positive controls in each run. Tissue cores incubated with phosphate-buffered saline lacking the specific primary antibody were used as negative controls.

Images of the tissue cores, obtained at 200× magnification with an HPF Nikon Eclipse E600 microscope, were used to determine the immunohistochemical H-scores. In brief, staining intensity was scored as none (0), weak (1), medium (2), or strong (3) in areas with tumor cells and the following formula was used to obtain H-scores ranging between 0 and 300: (0 × % tumor area with no staining) + (1 × % tumor area with weak staining) + (2 × % tumor area with medium staining) + (3 × % tumor area with strong staining). The H-score method is used to semiquantitatively determine the expression levels of specific proteins within tumor tissues and is a valuable, well-recognized, and validated method to assess biomarkers in human tissues [39]. H-scoring can be conducted for multiple proteins and enables comparison of the expression levels of different proteins because immunohistochemical staining for all proteins is simultaneously performed using a single tissue microarray. This approach minimizes the day-to-day experimental variation; only one optimized antibody is used for each core, and the performance of one antibody does not depend on the success of other antibodies [39].

Each core was individually assessed by two investigators, including one specialist histopathologist, who were blinded to the clinical data, and a consensus was reached. The average H-scores were generated by calculating the mean H-score of three cores in 187 patients, including 72 and 115 patients in the PDAC and distal bile duct/ampullary carcinoma cohorts, respectively, and by calculating the mean H-score of two cores in the remaining 43 patients, including 13 and 30 patients in the PDAC and distal bile duct/ampullary carcinoma cohorts, respectively.

### 2.4. Statistical Analysis

Associations between the expression levels of specific proteins were assessed using Spearman’s rank correlation coefficient. Cutoff H-scores, used for the stratification of cases according to the protein expression, were determined using X-tile software version 3.6.1 (Yale School of Medicine, USA) prior to statistical analyses [40]. Additionally, X-Tile software was used to identify 60 as the optimal age cutoff for stratifying patients based on clinical outcomes. The relationship between high and low protein expression and clinicopathological variables was assessed using the chi-squared test of association. Survival curves were plotted using the Kaplan–Meier method, and significance was determined using the log-rank test. Multivariate survival analysis was performed using the Cox proportional hazards regression model. All differences were deemed statistically significant at a *p*-value of < 0.05. All statistical analyses were performed using SPSS 23.0 software (IBM, Armonk, NY, USA).

## 3. Results

Antibody specificity was determined via Western blotting before immunohistochemical staining (Appendix A). Figure 1 shows representative photomicrographs of different staining patterns, i.e., weak, moderate, and strong cytoplasmic and nuclear immunohistochemical staining, for specific Trx system proteins in PDAC tissue microarrays. Appendix A shows representative photomicrographs of staining patterns for Trx system proteins in tissue microarrays of carcinomas of the pancreas, distal bile duct, and ampulla.

In the PDAC cohort, the median H-scores were 210 (range, 100–300), 225 (range, 50–300), 150 (range, 0–267), 75 (range, 0–225), and 75 (range, 0–225) for cytoplasmic Trx, nuclear Trx, cytoplasmic TxNIP, cytoplasmic TrxR, and nuclear TrxR, respectively. X-tile generated cutoff H-scores of 160, 234, 217, 58, and 58 for cytoplasmic Trx, nuclear Trx, cytoplasmic TxNIP, cytoplasmic TrxR, and nuclear TrxR, respectively. Based on these scores, high protein expression levels of 82.0% (61/74), 45.2% (33/73), 17.5% (14/80), 61.8% (47/76), and 61.8% (47/76) were observed for cytoplasmic Trx, nuclear Trx, cytoplasmic TxNIP, cytoplasmic TrxR, and nuclear TrxR, respectively.

In the distal bile duct/ampullary carcinoma cohort, the median H-scores were 150 (range, 0–300), 166.7 (range, 0–300), 166.7 (range, 0–300), 66.7 (range, 0–250), and 66.7 (range, 0–250) for cytoplasmic Trx, nuclear Trx, cytoplasmic TxNIP, cytoplasmic TrxR, and nuclear TrxR, respectively. X-tile generated cutoff H-scores of 142, 133, 85, 167, and 75 for cytoplasmic Trx, nuclear Trx, cytoplasmic TxNIP, cytoplasmic TrxR, and nuclear TrxR, respectively. Based on these scores, high protein expression levels of 60.2% (77/128), 63.3% (81/128), 75.6% (99/131), 10.7% (14/131), and 47.4% (63/133) were observed for cytoplasmic Trx, nuclear Trx, cytoplasmic TxNIP, cytoplasmic TrxR, and nuclear TrxR, respectively.

Spearman’s rank correlation coefficient analysis revealed that cytoplasmic TxNIP protein expression was statistically significantly, albeit weakly, correlated with cytoplasmic TrxR (r = 0.234, *p* = 0.038) and nuclear TrxR (r = 0.241, *p* = 0.032) protein expression in the PDAC cohort. In addition, cytoplasmic TrxR protein expression was strongly correlated with nuclear TrxR protein expression (r = 0.711, *p* < 0.001), and nuclear Trx protein expression was significantly correlated with cytoplasmic Trx protein expression (r = 0.549, *p* < 0.001). In the distal bile duct/ampullary carcinoma cohort, cytoplasmic Trx protein expression was significantly correlated with nuclear Trx (r = 0.653, *p* < 0.001), cytoplasmic TrxR (r = 0.436, *p* < 0.001), and nuclear TrxR (r = 0.328, *p* < 0.001) protein expression. In addition, nuclear Trx protein expression was significantly correlated with cytoplasmic TrxR (r = 0.2, *p* = 0.25) and nuclear TrxR (r = 0.376, *p* < 0.001) protein expression, and cytoplasmic TrxR protein expression was strongly correlated with nuclear TrxR protein expression (r = 0.653, *p* < 0.001).

### 3.1. Association between the Expression of Trx System Proteins and Clinicopathological Features

Appendix A show the associations between the expression of Trx system proteins and clinicopathological features in the PDAC and distal bile duct/ampullary carcinoma cohorts. In the PDAC cohort, the only significant association was between high cytoplasmic TrxR expression and lymph node metastasis (χ^2^ = 4.533, df = 1, *p* = 0.033). In the distal bile duct/ampullary carcinoma cohort, high cytoplasmic TxNIP and nuclear TrxR expression levels were significantly associated with age > 60 years (χ^2^ = 3.892, df = 1, *p* = 0.049 and χ^2^ = 5.091, df = 1, *p* = 0.024, respectively) (Table 2 and Table 3). Low TxNIP expression was significantly associated with the presence of perineural invasion (χ^2^ = 5.044, df = 1, *p* = 0.025) (Table 2). Furthermore, high nuclear TrxR expression was significantly associated with the presence of vascular invasion (χ^2^ = 10.548, df = 1, *p* = 0.001) and perineural invasion (χ^2^ = 5.314, df = 1, *p* = 0.021) (Table 3).

### 3.2. Relationship between the Expression of Trx System Proteins and Clinical Outcome

In the PDAC cohort, high cytoplasmic and nuclear Trx expression levels were significantly associated with better overall survival (*p* = 0.018 and *p* = 0.006, respectively) (Figure 2a,b). The expression levels of cytoplasmic TrxR, nuclear TrxR, and cytoplasmic TxNIP were not associated with overall survival (Figure 2c–e).

A multivariate Cox regression analysis was conducted by including potentially confounding factors such as sex, age, tumor size, tumor grade, tumor stage, lymph node status, and perineural and vascular invasion; however, none of these variables were independently associated with survival based on individual Kaplan–Meier analyses (*p* = 0.380, *p* = 0.694, *p* = 0.419, *p* = 0.820, *p* = 0.349, *p* = 0.063, *p* = 0.163, and *p* = 0.491, respectively). Nuclear Trx expression remained significant for survival based on the multivariate analysis (hazard ratio [HR] 0.316, 95% confidence interval [CI] 0.174–0.573; *p* < 0.001), whereas cytoplasmic Trx expression did not exhibit independent significance (HR 0.5, 95% CI 0.218–1.146; *p* = 0.102) (Table 4, panels A and B).

In the distal bile duct/ampullary carcinoma cohort, cytoplasmic Trx, nuclear Trx, cytoplasmic TrxR, and nuclear TrxR expression levels were not associated with overall survival (Figure 3a–d). However, high cytoplasmic TxNIP expression was significantly associated with better overall survival (*p* = 0.0002) (Figure 3e), which remained significant in the multivariate Cox regression analysis (HR 0.548, 95%CI 0.340–0.882; *p* = 0.013) (Table 4, panel C). In this cohort, the multivariate Cox regression analysis included tumor grade and stage, lymph node status, and perineural and vascular invasion, which were significantly associated with survival based on individual Kaplan–Meier analysis (*p* = 0.011, *p* = 0.004, *p* = 0.003 *p* = 0.001, and *p* = 0.012, respectively).

Trx and TrxR are expressed in both the nucleus and cytoplasm; therefore, we also analyzed the association between the expression levels of Trx system proteins with survival in patients with low nuclear/low cytoplasmic, low nuclear/high cytoplasmic, high nuclear/low cytoplasmic, and high nuclear/high cytoplasmic expression levels. The survival analysis of the PDAC cohort based on this categorization revealed no significant association between overall survival and the nuclear and cytoplasmic expression profiles of Trx or TrxR. Similarly, in the distal bile duct/ampullary carcinoma cohort, no significant association was observed between overall survival and the combined nuclear/cytoplasmic TrxR expression profile. However, overall survival was significantly longer in patients with low nuclear/high cytoplasmic Trx expression (n = 14) than in those in the other three subgroups (n = 114) when the analysis was conducted with each of the three subgroups evaluated separately (*p* = 0.017) (Figure 3f) or in combination (*p* = 0.002) (Figure 3g).

## 4. Discussion

Lower survival among patients with PDAC than among those with bile duct or ampullary cancer (7.8% vs. 29–37% and 41–45%, respectively) [2,3,4,5] may be partially explained by differences in tumor resectability rates (10–20% vs. 49.4% and 77.7%, respectively) [2,5,41,42,43]. In the present study, high cytoplasmic TrxR expression was associated with lymph node metastasis (*p* = 0.033) in patients with PDAC. A previous study involving 50 patients with oral squamous cell carcinoma also reported an association between low TrxR expression and lymph node metastasis (*p* = 0.027), albeit in a different cancer type [43]. Such data suggest that TrxR plays a role in the regulation of lymph node metastasis, which could be explored in future studies. Current analyses also indicated that high nuclear TrxR expression was significantly associated with the presence of vascular (*p* = 0.001) and perineural (*p* = 0.021) invasion, further suggesting a role of TrxR in tumor invasion in patients with cancers of the pancreas, distal bile duct, and ampulla.

The current data also revealed that low cytoplasmic TxNIP expression was significantly associated with the presence of perineural invasion (*p* = 0.025) in the distal bile duct/ampullary carcinoma cohort; this finding was consistent with that of a previous study reporting a significant association between high TxNIP expression and the absence of perineural invasion (*p* = 0.030) in 140 patients with gastroesophageal adenocarcinoma [28].

Furthermore, high cytoplasmic and nuclear Trx expression levels were significantly associated with better overall survival in patients with PDAC, and nuclear Trx expression remained significantly associated with survival based on multivariate Cox regression analysis (*p* < 0.0001). This finding, in light of expression in other tumor types, is somewhat unexpected. Raffle et al. observed that high Trx expression was significantly associated with poor overall survival (*p* = 0.004) in 12 patients with colorectal cancer [44]. In another study including 154 patients with ovarian cancer, low cytoplasmic Trx expression was significantly associated with better progression-free survival (*p* = 0.032), whereas nuclear Trx expression was not (*p* = 0.455) [25]. In a cohort of 65 patients with gastric cancer, high Trx expression was significantly associated with poor recurrence-free survival (*p* = 0.008) and overall survival (*p* = 0.015) [45]. In a recent study, high cytoplasmic and nuclear Trx expression levels were associated with adverse overall survival (*p* = 0.033 and *p* = 0.007, respectively) in 114 patients with medulloblastoma [46]. In the same study, high cytoplasmic Trx expression, but not high nuclear Trx expression, was significantly associated with adverse overall survival (*p* = 0.007) in 137 pediatric patients with high-grade glioma [46]. Notably, in the same study, high cytoplasmic and nuclear Trx expression levels were significantly associated with improved survival in 126 pediatric patients with low-grade glioma (*p* < 0.001 and *p* = 0.044, respectively) [46]. Furthermore, a study on 174 patients with Hodgkin’s lymphoma reported that high cytoplasmic Trx expression was significantly associated with better failure-free survival (*p* = 0.049), which remained significant in multivariate Cox regression analysis (*p* = 0.023) [47]. Conversely, three other studies showed no association between Trx protein expression and survival in patients with breast cancer, glioblastoma multiforme, and non-small cell lung cancer [27,46,48]. Therefore, the significance of Trx expression in patient survival may vary depending on the cancer type, hindering the generalization of the current findings.

In the present study, high Trx expression was identified as a good prognostic marker in patients with PDAC; this finding was in contrast to that of several in vitro studies, which suggest the utility of therapeutically targeting the Trx/TrxR system in cancer cells alone or in combination with chemotherapy or radiotherapy [12,23,49,50,51,52]. The contradictory findings between these in vitro studies and the present study based on immunohistochemical analysis may be explained by the limitation of the antibodies used here for the immunohistochemical detection of the oxidized or reduced states of Trx proteins. Nonetheless, our data may partially explain the disappointing clinical trials targeting the Trx system in PC [53].

We also found no association between the cytoplasmic and nuclear expression levels of TrxR or TxNIP and overall survival in patients with PDAC. This finding was in contrast with that of a study by Woolston et al., which included 98 patients with locally advanced breast cancer; the authors found that high expression levels of TrxR (*p* = 0.021) and TxNIP (*p* = 0.021) were significantly associated with improved distant metastasis-free survival [27]. Another study demonstrated that high cytoplasmic TrxR expression was significantly associated with adverse overall survival in patients with glioblastoma (*p* = 0.027), low-grade glioma (*p* = 0.027), high-grade glioma (*p* = 0.027), and medulloblastoma (*p* = 0.027) and that nuclear TrxR and TxNIP expression levels were associated with improved overall survival (*p* = 0.033 and *p* = 0.007, respectively) only in patients with low-grade glioma [44].

In the distal bile duct/ampullary carcinoma cohort, we observed a strong association between high cytoplasmic TxNIP expression and better overall survival, which remained significant in the multivariate Cox regression analysis. This finding was consistent with that of the abovementioned study by Woolston et al., which reported that high TxNIP expression was significantly associated with distant metastasis-free survival (*p* = 0.021) and overall survival (*p* = 0.037) in patients with breast cancer [27]. Lim et al. also observed a significant association between high TxNIP expression and longer relapse-free survival (*p* = 0.036) in 65 patients with gastric cancer [43]. Furthermore, high TxNIP expression was significantly associated with better disease-specific survival (*p* = 0.016) in 66 patients with gastroesophageal adenocarcinoma [28].

Further analysis of the distal bile duct/ampullary carcinoma cohort categorized according to the combined cytoplasmic and nuclear Trx expression status revealed that low nuclear/high cytoplasmic Trx expression was associated with longer overall survival than other three subgroups, regardless of whether they were evaluated separately or in combination. In contrast to the current findings, a study investigating the association between Trx expression and survival in patients with ovarian cancer found that high nuclear/low cytoplasmic Trx expression was associated with significantly better overall and progression-free survival compared with the other three groups, regardless of whether they were evaluated separately or in combination [25].

In summary, the present study demonstrates the importance of the expression levels of Trx system proteins in pancreatic, bile duct, and ampullary cancers. Our analyses revealed that high nuclear and cytoplasmic Trx expression levels were associated with better overall survival in patients with PDAC, high cytoplasmic TxNIP expression was associated with better survival in patients with bile duct or ampullary cancer, and these proteins were potentially important independent prognostic factors. The main limitation of the current study lies in the retrospective nature and the detection methods of Trx proteins, particularly the inability to distinguish between the oxidized and reduced states of Trx proteins. Additionally, some patient information is absent, including smoking and alcohol history, BMI/BRI status, and data on GSH/GSSG and p38/MAPK levels, which are proteins associated with the Trx system’s pathway.

These findings warrant independent follow-up studies with larger cohorts, studying the level of proteins associated with the Trx system’s pathway, such as GSH/GSSG and p38/MAPK levels, and employing more robust quantification techniques beyond IHC, such as qRT-PCR and ELISA, to strengthen the reliability and depth of our findings.

## 5. Conclusions

Trx system protein expression is important in pancreatic, distal bile duct, and ampullary cancers. The altered expression of certain Trx family proteins is potentially involved in the progression of periampullary cancers. The current findings warrant larger follow-up studies with larger cohort sizes to improve the accuracy and depth of current conclusions.

## Figures and Tables

**Figure 1 diseases-12-00227-f001:**
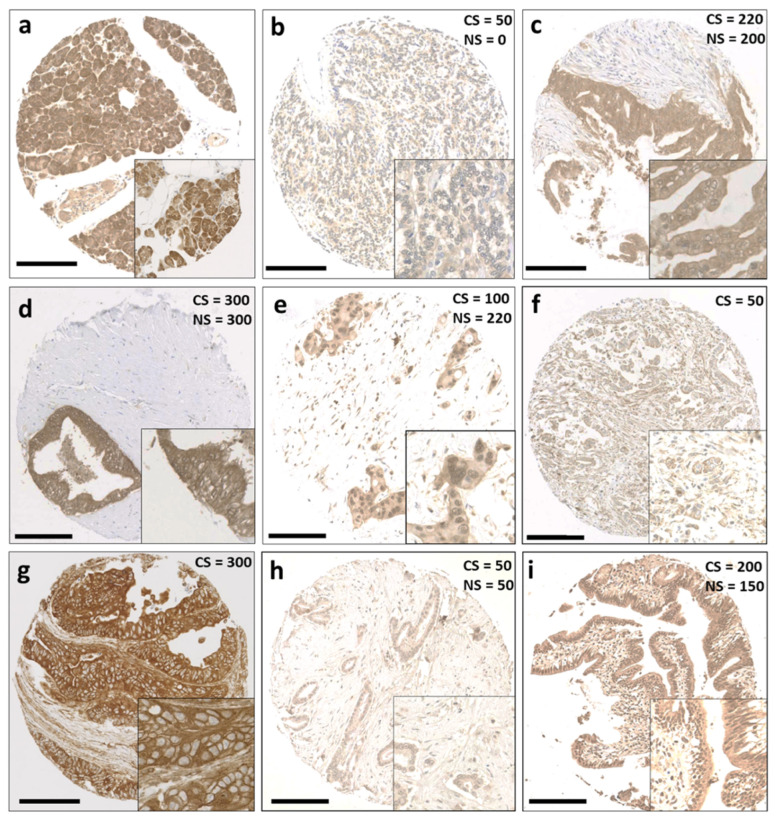
Representative staining patterns of Trx system protein expression in patients with pancreatic ductal adenocarcinoma. (**a**) Benign pancreatic tissue, (**b**) weak staining of Trx, (**c**) moderate staining of Trx, (**d**) strong staining of Trx, (**e**) weak cytoplasmic staining and moderate nuclear staining of Trx, (**f**) weak cytoplasmic staining of TxNIP, (**g**) strong cytoplasmic staining of TxNIP, (**h**) weak staining of TrxR, and (**i**) moderate staining of TrxR. Photomicrographs are at 10× magnification, with 20× magnification inset box. Scale bar: 200 μm. Abbreviations: CS = cytoplasmic H-score, NS = nuclear H-score.

**Figure 2 diseases-12-00227-f002:**
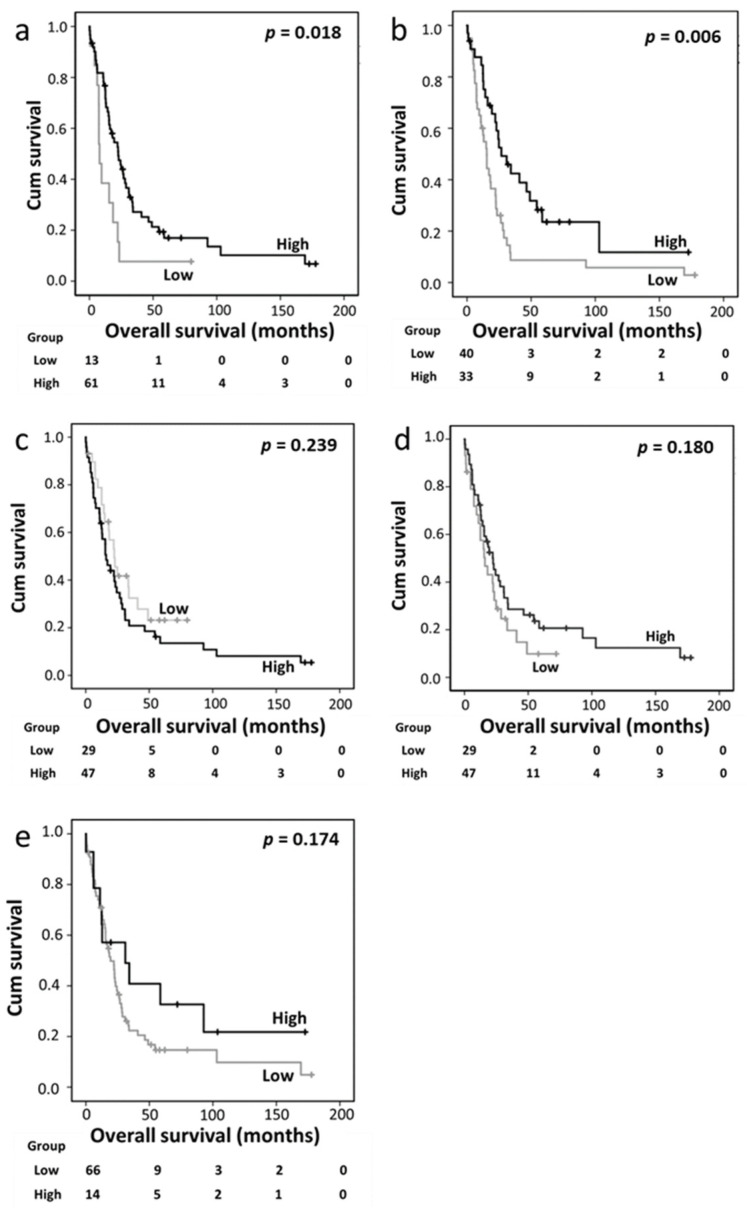
Kaplan–Meier analysis of overall survival showing the impact of cytoplasmic Trx (**a**), nuclear Trx (**b**), cytoplasmic TrxR (**c**), nuclear TrxR (**d**), and cytoplasmic TxNIP (**e**) expression in the pancreatic ductal adenocarcinoma cohort; significance was determined using the log-rank test.

**Figure 3 diseases-12-00227-f003:**
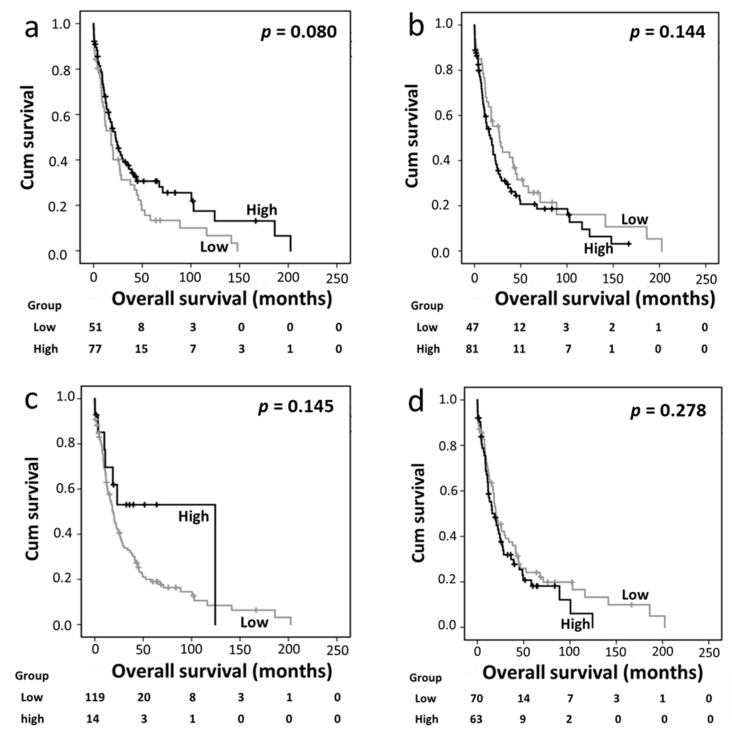
Kaplan–Meier analysis of overall survival in the distal bile duct and ampullary carcinoma cohort. Panels a–e show the impact of cytoplasmic Trx (**a**), nuclear Trx (**b**), cytoplasmic TrxR (**c**), nuclear TrxR (**d**), and cytoplasmic TxNIP (**e**) expression in the distal bile duct and ampullary carcinoma cohort; significance was determined using the log-rank test. The numbers shown below the Kaplan–Meier survival curves are the number of patients at risk in a particular month. Panels f and g show the overall survival analysis between different subgroups of Trx expression based on expression profiles. Low nuclear/high cytoplasmic expression showed longer overall survival than the other three subgroups either against each separate subgroup (**f**) or when the three subgroups were combined (**g**).

**Table 1 diseases-12-00227-t001:** Clinicopathological variables of 85 patients with pancreatic ductal adenocarcinoma (PDAC) and 145 patients with distal bile duct and ampullary cancers.

Characteristic	PDAC Cohort (n = 85)	Distal Bile Duct and Ampullary Cancer Cohort (n = 145)
	Frequency (%)	Frequency (%)
Age		
≤60 years	27 (31.8)	46 (31.7)
>60 years	56 (65.9)	98 (67.6)
Sex		
Male	52 (61.2)	81 (55.9)
Female	33 (38.8)	64 (44.1)
Tumor size		
≤2 cm	6 (7.1)	56 (38.6)
>2 cm	77 (90.6)	87 (60)
Tumor stage		
1	1 (1.2)	3 (2.1)
2	18 (21.2)	30 (20.7)
3	62 (72.9)	106 (73.1)
4	3 (3.5)	5 (3.4)
Lymph node stage		
Negative	28 (32.9)	52 (35.9)
Positive	54 (63.5)	85 (58.6)
Vascular invasion		
Absent	30 (35.3)	55 (37.9)
Present	54 (63.5)	88 (60.7)
Perineural invasion		
Absent	15 (17.6)	60 (41.4)
Present	69 (81.2)	84 (57.9)

**Table 2 diseases-12-00227-t002:** Associations between Trx and TxNIP protein expression and clinicopathological variables in the distal bile duct and ampullary carcinoma cohort.

Variable	Trx (Cytoplasmic)			Trx (Nuclear)			TxNIP		
	Low	High	*p*-Value	Low	High	*p*-Value	Low	High	*p*-Value
Age									
≤60 years	17 (13.1)	25 (19.2)	0.865	18 (13.8)	24 (18.5)	0.401	16 (11.9)	30 (22.4)	0.049 *
>60 years	37 (28.5)	51 (39.2)		31 (23.8)	57 (43.8)		17 (12.7)	71 (53.0)	
Sex									
Male	26 (19.8)	47 (35.9)	0.144	25 (19.1)	48 (36.6)	0.402	17 (12.6)	57 (42.2)	0.514
Female	28 (21.4)	30 (22.9)		24 (18.3)	34 (26.0)		17 (12.6)	44 (32.6)	
Tumor size									
≤2 cm	16 (12.3)	32 (24.6)	0.187	18 (13.8)	30 (23.1)	0.917	11 (8.20)	39 (29.1)	0.489
>2 cm	37 (28.5)	45 (34.6)		30 (23.1)	52 (40.0)		23 (17.2)	61 (45.5)	
Tumor stage									
1	1 (0.8)	2 (1.5)	0.778	2 (1.5)	1 (0.8)	0.329	0 (0.0)	3 (2.2)	0.497
2	11 (8.4)	16 (12.2)		12 (9.2)	15 (11.5)		5 (3.7)	23 (17.0)	
3	41 (31.3)	55 (42.0)		32 (24.4)	64 (48.9)		28 (20.7)	71 (52.6)	
4	1 (0.8)	4 (3.1)		3 (2.3)	2 (1.5)		1 (0.7)	4 (3.0)	
Node status									
Negative	17 (13.5)	29 (23.0)	0.456	17 (13.5)	29 (23.0)	0.937	27 (20.9)	22 (17.1)	0.573
Positive	35 (27.8)	45 (35.7)		29 (23.0)	51 (40.5)		40 (31.0)	40 (31.0)	
Vascular invasion									
Absent	21 (16.2)	29 (22.3)	0.821	22 (16.9)	28 (21.5)	0.186	11 (8.2)	41 (30.6)	0.371
Present	32 (24.6)	48 (36.9)		26 (20.0)	54 (41.5)		23 (17.2)	59 (44.0)	
Perineural invasion									
Absent	21 (16.0)	36 (27.5)	0.371	26 (19.8)	31 (23.7)	0.088	9 (6.7)	49 (36.3)	0.025 *
Present	33 (25.2)	41 (31.3)		23 (17.6)	51 (38.9)		25 (18.5)	52 (38.5)	

The number of observations for the cohort is shown for each clinicopathological variable; the table does not include the number of observations where clinicopathological data were not available. The frequency of observed clinicopathological variables is noted next to the variable subgroup. The *p*-values were calculated using Pearson chi-square test of association (χ^2^) or Fisher’s exact test in a 2 × 2 table if a cell count was less than 5. Significant *p*-values are indicated by *.

**Table 3 diseases-12-00227-t003:** Associations between TrxR protein expression and clinicopathological variables in the distal bile duct and ampullary carcinoma cohort.

Variable	TrxR (Cytoplasmic)			TrxR (Nuclear)		
	Low	High	*p*-Value	Low	High	*p*-Value
Age						
≤60 years	41 (30.6)	4 (3.00)	0.773	30 (22.4)	15 (11.2)	0.024 *
>60 years	79 (59.0)	10 (7.50)		41 (30.6)	48 (35.8)	
Sex						
Male	68 (50.4)	8 (5.90)	0.946	37 (27.4)	39 (28.9)	0.219
Female	53 (39.3)	6 (4.40)		35 (25.9)	24 (17.8)	
Tumor size						
≤2 cm	47 (35.1)	5 (3.70)	0.802	30 (22.4)	22 (16.4)	0.385
>2 cm	73.1 (54.5)	9 (6.70)		41 (30.6)	41 (30.6)	
Tumor stage						
1	3 (2.2)	0 (0.0)	0.100	2 (1.5)	1 (0.7)	0.461
2	23 (17.0)	4 (3.0)		14 (10.4)	13 (9.6)	
3	92 (68.1)	8 (5.9)		55 (40.7)	45 (33.3)	
4	3 (2.2)	2 (1.5)		1 (0.7)	4 (3.0)	
Node status						
Negative	9 (7.0)	38 (29.5)	0.326	43 (33.3)	6 (4.7)	0.522
Positive	22 (17.1)	60 (46.5)		73 (56.6)	7 (5.4)	
Vascular invasion						
Absent	44 (32.8)	5 (3.7)	0.944	35 (26.1)	14 (10.4)	0.001 *
Present	76 (56.7)	9 (6.7)		36 (26.9)	49 (36.3)	
Perineural invasion						
Absent	48 (36.5)	9 (6.7)	0.077	37 (27.4)	20 (14.8)	0.021 *
Present	73 (54.1)	5 (3.7)		35 (25.9)	43 (31.9)	

The number of observations for the cohort is shown for each clinicopathological variable; the table does not include the number of observations where clinicopathological data were not available. The frequency of observed clinicopathological variables is noted next to the variable subgroup. The *p*-values were calculated using Pearson chi-square test of association (χ^2^) or Fisher’s exact test in a 2 × 2 table if a cell count was less than 5. Significant *p*-values are indicated by *.

**Table 4 diseases-12-00227-t004:** Multivariate Cox regression analysis of factors associated with overall survival in patients with pancreatic ductal adenocarcinoma (PDAC), distal bile duct cancer, and ampullary cancer.

A	PDAC cohort	*p*-value	EXP (B)	95% CI for EXP (B)	
				Lower	Upper
	Cytoplasmic Trx expression	0.102	0.5	0.218	1.146
	Sex	0.579	1.178	0.662	2.096
	T stage	0.493	0.772	0.368	1.619
	Node status	0.113	1.727	0.878	3.398
	Vascular invasion	0.833	1.068	0.581	1.962
	Perineural invasion	0.305	1.521	0.682	3.392
	Tumor size	0.221	1.825	0.697	4.784
	Patient age	0.804	0.929	0.518	1.666
	Grade	0.874	1.045	0.608	1.796
B	PDAC cohort	*p*-value	EXP (B)	95% CI for EXP (B)	
				Lower	Upper
	Nuclear Trx expression	<0.0001 *	0.324	0.177	0.592
	Sex	0.182	1.482	0.831	2.64
	T stage	0.99	1.004	0.497	2.031
	Node status	0.015	2.353	1.178	4.698
	Vascular invasion	0.292	1.412	0.744	2.679
	Perineural invasion	0.985	1.008	0.458	2.217
	Tumor size	0.132	2.202	0.788	6.155
	Patient age	0.601	1.176	0.642	2.154
	Grade	0.644	1.135	0.664	1.94
C	Distal bile duct and ampullary cancer cohort	*p*-value	EXP (B)	95% CI for EXP (B)	
				Lower	Upper
	TxNIP expression	0.013 *	0.548	0.34	0.882
	T stage	0.063	1.634	0.974	2.741
	Node status	0.054	1.586	0.992	2.537
	Vascular invasion	0.916	0.975	0.606	1.568
	Perineural invasion	0.251	1.309	0.827	2.073
	Grade	0.05	1.5	0.999	2.251

EXP (B) is used to denote hazard ratio, and 95% CI is used to denote 95% confidence interval. Significant *p*-values are indicated by *.

## Data Availability

Additional data were submitted as Appendix A with original H-scores and images held in a secure database. Such information can be made available upon request to the corresponding author.

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
