# Peer review of "Thioredoxin System Protein Expression in Carcinomas of the Pancreas, Distal Bile Duct, and Ampulla in the United Kingdom"

_diseases, 2024, doi:10.3390/diseases12100227_

Round 1
Reviewer 1 Report
Comments and Suggestions for Authors
Attached please see my comments.
Manuscript ID: am-2024-07528d “Thioredoxin system protein expression in carcinomas of the pancreas, distal bile duct, and ampulla”
In this paper, the authors retrospectively explored the expression of the following redox proteins in the distal bile duct, the ampullary carcinoma, and the pancreatic tissues via the immunohistochemistry: thioredoxin (Trx), thioredoxin-interacting protein (TxNIP), and thioredoxin reductase (TrxR). They studied the expression levels of the proteins over the tissue samples collected from the patients with pancreatic ductal adenocarcinoma
(PDAC), distal bile duct, or ampullary carcinoma. They concluded that the expression levels of the Trx and nuclear Trx in ampullary, pancreatic, and distal bile duct cancers. The subject is interesting and is suitable for the Diseases. However, I would recommend the authors to incorporate the following comments before this paper can be considered for publication in Diseases.
I strongly suggest the authors to consider multiple techniques alongside the
immunohistochemistry to verify their findings.
The quantitative reverse transcription polymerase chain reaction (qRT-PCR) by measuring the gene expression levels of the Trx, TxNIP, and TrxR through quantification of their mRNA levels in the cells of the tissues. Another technique which needs to be considered for this study is the enzyme linked immunosorbent assay (ELISA).
These two techniques will provide very precise quantitative measurements of the proteins of the interest in the tissue samples. The immunohistochemistry does not provide a detailed information regarding the levels of expressions of the target proteins and therefore it might not be the sufficient technique to rely for making such significant conclusion as it was made by the authors for this study.
Reviewer 2 Report
Comments and Suggestions for Authors
Comments:
1. Are all patients from UK? If yes, please add ".... in UK" in the title.
2. What is the main limitation of the current study? please discuss.
3. Since TrxR proteins are associated with GSH/GSSG, what are patients' GSH/GSSG levels?
4. Since p38/MAPK pathway is associated with TrxR proteins, what are patients' p38/MAPK levels?
5. Please explain why age cutoff is 60?
6. Which lymph node is used for stage?
7. Please add patients' smoking and alcohol history and BMI/BRI.
8. Please provide a schematic cartoon for summary.
Round 2
Reviewer 1 Report
Comments and Suggestions for Authors
The revised manuscript is acceptable for publication in Diseases.
Reviewer 2 Report
Comments and Suggestions for Authors
My questions were answered. No more comments.